# Feasibility Study for the Remanufacturing of H13 Steel Heat-Treated TBM Disc Cutter Rings with Uniform Wear Failure Using GMAW

**DOI:** 10.3390/ma16031093

**Published:** 2023-01-27

**Authors:** Kui Zhang, Shuhao Dai, Boyan Jiang, Xuejun Zheng, Jingang Liu, Xuhui Zhang

**Affiliations:** 1Postdoctoral Research Station for Mechanics, Xiangtan University, Xiangtan 411105, China; 2School of Mechanical Engineering, Xiangtan University, Xiangtan 411105, China; 3Engineering Research Center of Complex Tracks Processing Technology and Equipment of Ministry of Education, Xiangtan University, Xiangtan 411105, China; 4School of Engineering and Design, Hunan Normal University, Changsha 410081, China

**Keywords:** TBM disc cutter, H13 steel cutter ring, heat treatment, remanufacturing, GMAW

## Abstract

Given that the heat treatment states of the base metal have a great influence on the surfacing repair layer, this paper carried out a feasibility study for the remanufacturing of the failed cutter rings of TBM disc cutters with uniform wear (hereinafter referred to as normally-worn ring) using the gas metal arc welding technology (GMAW). Firstly, this paper developed a heat treatment process route for H13 steel cutter rings. Secondly, the heat treatment process is numerically analyzed based on the developed route, and the rationality of the route is verified from the distribution characteristics of temperature, phase, and stress fields. Subsequently, heat treatment tests were carried out, and the physical and mechanical properties of the base metal samples prepared under laboratory conditions were evaluated respectively and systematically. Based on the comprehensive performance evaluation value calculated by the weighted comparative analysis method, it was clear that the comprehensive performance of the quenched base metal samples was 7.6% higher than that of the engineering cutter ring interior. Therefore, it is reasonable to replace the failed engineering cutter rings repaired under laboratory conditions with the prepared samples as economical alternatives. Finally, the remanufacturing of the base metal samples using GMAW was carried out, and then the remanufacturing performance of the base metal samples was analyzed. The study concluded that the comprehensive performance of the surfacing repair layer was slightly lower than that of the engineering cutter ring edge (4.1%), thus proving that the idea of surfacing remanufacturing of the normally-worn ring proposed in this paper was basically feasible.

## 1. Introduction

As modern construction equipment, the full-face tunnel boring machine (TBM) integrates mechanical, hydraulic, electronic, control, and laser technologies [1]. Compared to the traditional drilling and blasting method (DBM), TBM is more efficient and safer. Therefore, it has been widely used in tunnel and underground space engineering [2,3]. During the construction of TBM tunnels, the TBM cutters are subjected to large thrust and torque from the cutterhead, causing the rock to peel off from the excavation face due to rock fractures originating from compressive stress concentrations and propagating approximately perpendicularly to the direction of maximum compression [4]. This method of rock-breaking is extremely efficient. However, due to the harsh working conditions of TBM tunnelling, it is straightforward to cause various failures, such as breakage of bearings, uniform and non-uniform wear of cutter rings, etc. [5]. According to the investigation by Fu et al. [6] on the cutter ring failure of the shield machine used in the Jinan (northeast China) subway tunnel construction, from April 2018 to May 2018, more than 80% of the cutter rings on the cutter head failed in various forms. Jin et al. [6] performed a probabilistic analysis of cutter failure and cutter group failure in hard rock excavation and proposed an adoptable reliability design chart for the industry to estimate the potential failure of the cutter. Remanufacturing machining has been recognized as an effective technology, which not only improves the energy and materials utilization, but also decreases environmental emissions and production costs [7]. Remanufacturing technology has been widely used in the automotive, aviation, and marine fields and has achieved good economic benefits [8,9,10]. In order to avoid direct scrapping of failed cutter rings and to reduce carbon emissions, He et al. [11] proposed a remanufacturing scheme for failed cutter rings of TBM disc cutters with uniform wear (hereinafter referred to as normally-worn rings) based on gas metal arc welding technology (GMAW). Initially, they verified the feasibility of remanufacturing repair of cutter rings that are made of AISI H13 hot work tool steel (hereinafter referred to as H13 steel). However, the heat treatment states of the base metal samples were not paid too much attention. Zhang et al. [12] showed that, when remanufacturing failed metal parts, the heat treatment state of the base metal samples can have a huge impact on the remanufacturing performance.

Under laboratory conditions, before economically and reasonably assessing the feasibility of remanufacturing failed cutter rings, the first task is to prepare the base metal samples that are close to the engineering reality. To ensure that the comprehensive performance of cutter rings satisfies the requirements of the geological conditions of the tunnelling stratum, the microstructures of the cutter rings can be properly regulated in engineering by planning the heat treatment process route. For example, since the cutters are normally subjected to heavy loads and impacts when tunneling in hard rock stratum, the heat treatment processes of quenching and tempering are usually used to prepare the cutter rings with high external hardness and optimal internal toughness. With the further increase of rock hardness, it is necessary to optimize the heat treatment process route and its parameters to improve the physical and mechanical properties of cutter rings (including the wear resistance, shear strength, impact toughness, hardness, etc.) [13]. In addition, the heat treatment process route should also be reasonably designed according to the material and chemical composition of the cutter rings. Bae et al. [14] discovered that, after a reasonable tempering treatment, H13 steel could achieve excellent hardness and toughness. Li et al. [15] investigated the effect of different heat treatment process parameters on the comprehensive performance of 42CrMo steel and then optimized the comprehensive performance of the low alloy steel. Jiang et al. [16] used different tempering temperatures to regulate the amount of precipitated phases and austenite phase transformation in C-Cr-Mo-V alloy steel for cutter rings to regulate its hardness and toughness. Since it is recommended that the cutter ring samples used to be repaired in remanufacturing tests have the same heat treatment state as the engineering cutters, the preparation of ring samples in standard size, especially that of reduced-scale samples for economic considerations, requires a lot of original research work due to the following reasons: On the one hand, as mentioned earlier, the heat treatment process route and its process parameters for engineering cutter rings need to be adaptively designed according to the geological conditions, the characteristics of the base metal samples, and the geometry of the cutter rings, and therefore there is no unified standard; on the other hand, the heat treatment process for engineering cutter rings made of H13 steel, which is the core technology secret owned by cutter ring manufacturing companies, has not been fully disclosed to the public. In addition, by referring to the existing heat treatment processes of the engineering cutters, it needs to be further verified whether it is feasible to prepare the reduced-scale samples.

Before reasonably assessing the feasibility of remanufacturing failed cutter rings, another key task is to put forward a comprehensive performance assessment method that matches the engineering reality for cutter rings. There are few comprehensive performance evaluation methods for cutter rings due to the fact that: (1) In general, the physical and mechanical properties of cutter rings used in different TBM tunneling projects vary greatly to cope with the different geological conditions; and (2) there are many performance evaluation indices for TBM cutters, such as indent depth, hardness, shear strength, and impact toughness; some of them are incompatible with each other while the others are compatible.

Considering that H13 steel is widely used in the manufacture of cutter rings [17], this paper takes the normally-worn ring as the object that will be repaired. Firstly, in order to prepare the base metal samples in the quenched state with similar physical and mechanical properties as engineering cutter rings, a heat treatment process route for H13 steel cutter rings was developed in this paper, and key parameters of the heat treatment process were then designed based on available information and the metal material calculation software JMatPro. Secondly, the finite element (FE) model of a 1/64 cutter ring was established using large-scale commercial welding FE analysis software Sysweld, and the whole process of heat treatment was simulated and analyzed according to the process route. Thirdly, heat treatment tests were carried out, in which the base metal samples were simplified to plate samples of the same material in order to reduce the test cost, and then based on the weighted comparative analysis method, a formula was proposed to calculate the comprehensive performance evaluation value Si of each sample prepared under laboratory conditions relative to different parts of the selected engineering cutter ring samples. Subsequently, by comparing the measured mechanical and physical properties of the engineering cutter ring, it was verified that the comprehensive performance of the base metal was similar to that of the interior zone of the engineering cutter ring. Finally, based on the established remanufacturing test platform with a six-degree-of-freedom welding robot, the remanufacturing tests of the base metal samples were carried out using GMAW. After that, the physical and mechanical properties of the surfacing layers were tested, the microstructures of the remanufactured samples were observed, and the comprehensive performance of the repair layers were also evaluated by the weighted comparative analysis method, which finally verified the feasibility of the surfacing remanufacturing of the heat-treated H13 steel cutter rings. In the past, the remanufacturing performance of cutter rings was usually evaluated empirically or by a single performance, but the weighted comparative analysis method proposed in this paper is relatively more scientific. The research results are of great significance for the application of cutter ring remanufacturing.

## 2. Simulation Analysis of the Heat Treatment Process

### 2.1. Formulation of the Heat Treatment Process Route

Currently, the industry-accepted heat treatment process for cutter rings consists of no less than twice of preheating and tempering and once quenching [18]. Comprehensive literature reports show that the preheating temperatures and their holding times were set to 650 °C for 90 min and 850 °C for 90 min, respectively. The thermal equilibrium phase diagram of H13 steel calculated by JMatPro is shown in Figure 1. It can be seen from the figure that: (1) At 834.71 °C, ferrite began to transform into austenite; (2) at 1122.92 °C, all austenite transformations were finally completed; and (3) at 1390 °C, some of the austenite starts to melt until 1480 °C, when the austenite melts completely. Thus, the quenching temperature was set to 1140 °C (between 1122 °C and 1390 °C) to ensure that the microstructure of the cutter ring was completely converted into austenite and was not melted. The continuous cooling transition curve (CCT) calculated by JMatPro (Sente Software Ltd., Surrey, UK) is shown in Figure 2. It can be seen from the figure that the martensite and bainite transformation temperatures, namely *M_s_* and *B_s_*, are 352.3 °C and 508.2 °C, respectively. Since the tempering temperature should be higher than *B_s_*, it was set to 540 °C.

The heating time is determined by the following two empirical formulas [19]: (1)t=(a+b+c)×K×D
where *t*—the heating time during the quenching process (min), *a*—the heating coefficient after the second preheating process (0.4 min/mm), *b*—the heating coefficient during the first preheating process (2 min/mm), *c*—the heating coefficient during the second preheating process (1 min/mm), *K*—the correction coefficient (1), and *D*—the effective thickness of workpieces (76 mm).
(2)T=A·D+B
where *T*—the heating time during the tempering process (min), *A*—the heating coefficient (2 min/mm), *D*—the effective thickness of workpieces (76 mm), and *B*—the additional time (10 min).

Using the above two formulas, the quenching heating time *t* and tempering heating time *T* were obtained as 270 and 180 min, respectively. The heat treatment process route of the H13 steel cutter ring is shown in Figure 3. As can be seen from Figure 4, the heat transfer coefficient of quenching oil is much higher than that of air, so the efficiency of oil quenching and cooling is higher than that of air cooling. Quenching oil was chosen as the medium for the cooling stage during the quenching process to increase the cooling rate; the air was chosen as the medium for the cooling stage during the tempering process to slow down the cooling rate and thus reduce the residual stress, where the room temperature was given as 20 °C.

### 2.2. Simulation Parameters and FE Model

By virtue of Sysweld combined with the simulation engineering framework toolset Visual-Environment (VE), a FE model was established to simulate the proposed heat treatment process and analyze the rationality of each parameter set for the heat treatment process route in terms of three aspects: temperature field, phase transition field, and stress field. The chemical composition of the simulated cutter ring is the same as that of the base metal samples shown in Table 1. As shown in Figure 5, firstly, in order to appropriately reduce the simulation scale, an FE model of 1/64 of the 17-inch cutter ring was established using the boundary division meshing method. The preferential division into two-dimensional units on the cutter ring surface, whose unit size was up to 1.5 mm, after generating the mesh on the first layer, gradually divided new units toward the cutter ring to create a total of 5 layers with a total thickness of 5 mm. The movement in the negative direction of the X-axis of the face of the cutter ring in contact with the ground is constrained, and the external surface of the cutter ring (i.e., the red surface in the figure) was set as the surface in contact with the air.

### 2.3. Simulation Analysis of Temperature Field

The simulated temperature change curves at the interior and exterior points of the cutter ring and the temperature difference curve between the interior and exterior points of the cutter ring are shown in Figure 6. It can be seen from the figure that the temperature change curves at the interior and exterior points of the cutter ring were basically consistent with the pattern expected from the developed heat treatment process route. The cooling speed at the exterior point of the cutter ring was faster than that at the interior; the temperature difference between the interior and exterior points reached 723.63 °C at 500.64 min (30,038.4 s); the temperature difference between the interior and exterior points was nearly 0 °C after the heat preservation process was reasonably set in the process of twice preheating, once quenching, and twice tempering. 

### 2.4. Simulation Analysis of Phase Transition Field

This section analyzed the laws of phase transformation with time, starting from austenite, bainite, and martensite, respectively.

8823 s (147 min)

The temperature and microstructure cloud diagrams at 8823 s (the beginning of the heat preservation stage after the second preheating process) are shown in Figure 7. Due to the slow temperature rise inside the cutter ring, it was evident from Figure 7a that there was a slight temperature difference between the interior and exterior points of the cutter ring; the minimum temperature at that moment was about 843 °C, which was slightly higher than the austenitization temperature of 834.71 °C calculated by JMatPro. Figure 7b shows that, at this time, a portion of the ferrite began to transform into austenite (not less than 9%). This was consistent with the austenitization temperature predicted by JMatPro. 

2.30,000 s (500 min)

The microstructure and temperature cloud diagrams at 30,000 s (the last second of the heat preservation stage during the quenching process) are shown in Figure 8. As shown in the figure, the temperatures at the cutter ring interior and exterior points had reached the quenching temperature (1140 °C), and the austenite content was nearly 100%, proving that the heat preservation time and temperature during the quenching process were set appropriately.

3.30,030 s (500 min) and 30,033 s (500.5 min)

The microstructure cloud diagrams at 30,030 s and 30,033 s (the initial stage of the cooling stage during the quenching process) are depicted in Figure 9. Figure 9 shows that 1.1% of bainite had formed at this time, indicating that the temperature had dropped below *B_s_* and some austenite had begun to change into bainite. Figure 10 shows that 1.6% of martensite had formed at this time, indicating that the temperature had dropped below *M_s_* and some austenite had begun to change into martensite.

4.30,045 s (500.85 min)

The microstructure and temperature cloud diagrams at 30,045 s (the initial cooling stage of the quenching process) are shown in Figure 10. It can be seen from Figure 10a–c that:(1)some zones inside the cutter ring had not been cooled below *M_s_*; since *B_s_* is greater than *M_s_*, the austenite in these zones had not been converted to martensite, resulting in a wider distribution of bainite than martensite in the cutter ring.(2)due to the high heat transfer coefficient of the quenching oil (see Figure 4), the cooling rate was rapid, resulting in the temperature remaining at *B_s_*~*M_s_* for a short time; additionally, because martensite forms extremely quickly, the maximum amount of martensite (43%) was higher than the maximum amount of bainite (36%).(3)there was still a large amount of austenite that had not undergone phase transformation at this moment, and the quenching process will continue for some time.(4)bainite and martensite were generated inward from the edge angle of the cutter ring.(5)since there were chamfers at the cutter ring edge, the heat exchange area of the cutter ring edge was smaller than that of the ring root, which resulted in a relatively slow phase transformation speed of the ring edge.

By comparing Figure 10d and Figure 2, it can be seen that the simulation results ($B_{s}=460.9\;{}^{\circ}C$, $M_{s}=336.3\;{}^{\circ}C$) were basically consistent with those predicted by JMatPro $(B_{s}=509.2\;{}^{\circ}C$, $M_{s}=352.3\;{}^{\circ}C$).

5.30,960 s (516 min)

The microstructure, temperature, and hardness cloud diagrams at 30,960 s (the last second of the cooling stage during the quenching process) are shown in Figure 11. From Figure 11d, it can be seen that the temperature of both the exterior and interior of the cutter ring had dropped to room temperature at this time, which indicates that the cooling time was set reasonably. From Figure 11a–c, it can be seen that austenite was essentially completely transformed, with bainite content ranging from 2%–38% and martensite content ranging from 58%–79%; the content of martensite gradually decreased from the exterior to the interior of the cutter ring, while the bainite content gradually increased from the exterior to the interior of the cutter ring; from Figure 11e, it can be seen that the hardness of martensite was higher than that of bainite, which led to the same distribution of hardness as that of martensite.

6.69,360 s (1156 min)

The microstructure, temperature, and hardness cloud diagrams at 69,360 s (the last second of the cooling stage during the second tempering process) are shown in Figure 12. As shown in Figure 4, the heat transfer coefficient of the air was low, which led to the slow cooling speed of the cutter. As can be seen from Figure 12c, the minimum temperature of the cutter ring at this moment was about 23 °C. It was slightly higher than the room temperature, which indicates that the time setting at the cooling stage of the tempering process was reasonable. As shown in Figure 11b and Figure 12a, the content of martensite before and after tempering was the same, indicating that the tempering process results in tissue transformation rather than phase transformation (after tempering, the structure was changed into secondary structures like tempered martensite, tempered troostite, and tempered solitaire). Comparing Figure 11e and Figure 12d, it can be seen that the hardness cloud diagram remained the same after tempering. Still, in actual production, the hardness of the cutter ring before tempering is normally higher than that after tempering due to the fact that the outer martensite is harder than its secondary structure. What should be noted was that the simulation did not consider the tissue transformation after tempering.

### 2.5. Simulation Analysis of Stress Field

30,000–30,960 s (approximately 500–516 min)

To facilitate the description, the start time of the cooling stage of the quenching process was set to 0 s (relative to the 30,000 s of the heat treatment process), and the ending time was set to 960 s (the 30,960 s of the heat treatment process). Figure 13 shows the change diagram of the residual stress at the interior and exterior points of the cutter ring in Figure 5 during the cooling stage. Combining the phase transition law obtained from the simulation in Section 2.4, it can be seen as follows:(1)0–59.03 s: during this period, the temperature inside and outside the cutter ring was higher than *B_s_*; significant transformation of austenite to bainite or martensite cannot be observed, only the effect of temperature change on stress variation needs to be considered. In addition, since the cooling rate of the interior point was faster than that of the exterior point, the surface of the cutter ring was in tension while the inside of the cutter ring was in compression. With the passage of time, the temperature difference between the interior and exterior points increased, and the tensile stress at the exterior point increased rapidly to 152.50 MPa, while the compressive stress at the interior point also rose rapidly to 19.57 MPa.(2)59.03–77.42 s: during this period, the temperature inside and outside the cutter ring progressively falls below *M_s_*; a gradual increase in the production of martensite and bainite on the surface of the cutter ring could be observed, combined with Figure 6, it can be seen that the temperature difference between the interior and exterior points of the cutter ring gradually decreased after 38.4 s, and the tensile stress on the surface of the cutter ring and the compressive stress within the ring also decreased. At the same time, the effect of phase transformation on residual stress also gradually increased. As the phase transformation expansion coefficients of martensite and bainite are much higher than the thermal expansion coefficient of H13 steel, the expansion induced by phase transformation made the exterior point change from a tensile to a compressive stress state. At this time, the phase transformation generated inside the cutter ring was almost negligible. Due to the expansion of the cutter ring surface, the compressive stress inside the cutter ring slowly decreased and then transformed into tensile stress.(3)77.42–83.17 s: With the elapse of time, the temperature inside and outside the cutter ring becomes lower and lower. On the one hand, a large amount of bainite was generated inside the cutter ring during the period, and the higher phase change expansion coefficient made the internal tensile stress decrease gradually; on the other hand, since the phase transformations on the surface of the cutter ring were almost complete, the stress variation of the cutter ring was mainly influenced by the internal phase change. The above two facts indicate that as the temperature dropped, the phase of the cutter ring transformed from the inside to the outside; during this period, the phase transformation in the subsurface layer reduced the compressive stress in the surface layer, causing the stress state of the exterior point to change to tensile stress eventually.(4)83.17–960 s: during this period, the phase transformation of the cutter ring had been basically completed, and the distribution of residual stress in the cutter ring was mainly affected by temperature. When combined with Figure 6, it is clear that, as the temperature difference decreased, the tensile stress at the exterior point gradually decreased and then transformed into compressive stress. When the cooling stage of the quenching process was over, the cutter ring was subjected to compressive stresses on the surface and tensile stresses within the ring.

2.69,360 s (1156 min)

This is the last second of the heat treatment process. Figure 14 shows the residual stress cloud at 30,960 s and 69,360 s. Comparing Figure 14a with Figure 14b, it can be seen that the distribution pattern of residual stress in the cutter ring before and after tempering was basically the same. The maximum tensile stress was reduced by 28.2%, while the maximum compressive stress by 28.9%, which indicated that the tempering processes can effectively reduce the residual stress.

## 3. Materials and Methods

### 3.1. Experimental Tasks

The experimental tasks were set as follows:

Task 1: engineering cutter ring sampling and performance testing to provide basic data for comparative analysis in Tasks 2 and 3.

Task 2: firstly, the H13 steel cutter ring samples in the annealed state were treated according to the heat treatment process route developed in Section 2.1. The resulting heat-treated samples were sampled and tested for performance. Secondly, assess the differences in physical and mechanical properties between the heat-treated samples and the engineering cutter rings selected in Task 1. Finally, the base metal samples, which have similar physical and mechanical properties to those of the engineering cutter ring, were prepared for Task 3.

Task 3: the base metal samples prepared in Task 2 were remanufactured using GMAW, and then the remanufactured samples were obtained. The remanufactured samples were sampled, and performance tests were conducted. The feasibility of surfacing remanufacturing H13 steel cutter rings in the quenched state was evaluated by comparing the physical and mechanical properties of the base metal samples obtained in Task 2 with those of the engineering cutter rings obtained in Task 1.

### 3.2. Test Platform and Test Materials

For Task 1, a certain type of 17-inch engineering cutter ring was selected as the test object, and its chemical composition is shown in Table 1.

For Task 2, According to the cutter ring heat treatment process route developed in Section 2.1, the base metal samples were prepared using a box-type resistance furnace as the main preparation apparatus. Considering the large size of the engineering cutter ring, in order to reduce the test cost and difficulty, the annealed cutter ring samples were simplified into plate samples with the dimensions of 150 mm × 50 mm × 10 mm which have a chemical composition similar to the engineering ring (see Table 1).

For Task 3, as shown in Figure 15, using the six-degree-of-freedom welding robot as the basic test platform, the cutter ring surfacing remanufacturing test was carried out, and then the remanufacturing performance was analyzed. The test platform includes MOTOMAN-GP8 robot (including a demonstrator), Yaskawa MOTOWELD-RD500 welder, and the control cabinet. Among them, the robot arm is equipped with a welding gun at the end, which can design a high-precision welding trajectory by means of the demonstrator. The control cabinet adjusts the welding current, voltage, and wire feed rate to control the parameters of the cutter ring surfacing remanufacturing process. The parameters of the surfacing remanufacturing process, such as the welding current, voltage, wire feed rate, etc., can be controlled using the control cabinet. In Task 3, the SWD-H13 flux-cored wire, which has good compatibility with H13 steel in mould repair, was selected. Its chemical composition is shown in Table 1. Table 2 describes the test projects and the test instruments for each task.

### 3.3. Sampling for Performance Tests

Figure 5 shows three sampling regions for the engineering cutter ring samples selected in Task 1, namely the edge, interior, and root. 

All samples used for performance testing in Task 2 and Task 3 were prepared according to Figure 16. As shown in Figure 16, metallographic observation samples, wear-resistant samples, and shear samples were mainly obtained by single-pass and single-layer surfacing to improve test efficiency; considering that the weld reinforcement of the samples obtained by single-pass and single-layer surfacing was too small to meet the impact testing requirements, the impact samples were prepared by multi-pass and multi-layer surfacing. In order to reduce the number of samples taken, the hardness test was performed on the backside of the impact samples before the impact test. The roughness of the friction plane Ra is 6.3 µm.

The shapes and dimensions of the samples prepared in Task 1 were the same as those of the samples in Tasks 2 and 3.

### 3.4. Performance Test Schemes

Wear resistance

The wear-resistant performance was characterized by indent depth *h* and hardness *H* [20]. The test procedure for measuring indent depth *h* was as follows: using CFT-I friction tester, dry friction test was firstly carried out in which a standard steel ball with a diameter of 4 mm repeatedly rubbed the friction plane of the wear-resistant sample; secondly, the indent depth *h* of the worn wear-resistant sampling was measured using VHX-2000C super depth-of-field microscope.

2.Shear strength

The shear strength *τ* of the shear sample was tested using WAW-300 universal testing machine. During the test, the shear loading speed was set to 2 mm/min, and the maximum applied shear load was set to 25 kN.

3.Impact toughness

The impact toughness *a* of the impact sample with a V-shaped fracture, as shown in Figure 16, was tested in accordance with ISO 148-1:2006. During the test, the impact angle was set to −150°, and the impact energy was set to 300 J.

The test schemes for observing microstructure and measuring hardness *H* were relatively simple and will not be presented here.

## 4. Experimental Results and Discussion

The physical and mechanical properties of the engineering cutter ring samples, the base metal samples, and the surfacing repair layer samples are shown in Figure 17. For each part of the cutter ring with the same chemical composition, the indent depth *h* was inversely proportional to the hardness *H*. While the cutter ring, base metal samples, and surfacing repair layer were all made of H13 steel, their chemical compositions were different. The different chemical compositions led to different compatibilities between the friction object and the steel pellets used for friction testing. As the base metal samples and steel pellets were well-compatible, the indent depth on the base metal samples was greater. For materials with the same chemical composition, hardness was inversely proportional to impact toughness [21]. The test results for hardness and impact toughness for different parts of the cutter ring verified the above conclusions.

### 4.1. Physical and Mechanical Properties of the Base Metal

If the radial wear of the cutter ring reaches 35 mm during engineering maintenance, the ring is considered failed [22]. At this time, the ring edge is worn away by the rock, and the inside of the cutter ring interior is exposed. In order to better match the engineering reality, the physical and mechanical properties of the base metal samples prepared in Task 2 should be as close as possible to those of the interior of the engineering cutter ring.

1.Single performance evaluation

It can be seen from Figure 17 that the engineering cutter ring edge had high wear resistance but relatively low impact toughness, while the cutter ring interior had high toughness but relatively low wear resistance, which basically met the requirements of the extreme working conditions in TBM tunnelling. 

Compared to the physical and mechanical properties of the engineering cutter ring samples sampled from each region, the base metal samples had both advantages and disadvantages. Therefore, it was not easy to determine which sample had the best performance. Only in terms of the number of advantageous items were the physical and mechanical properties of the base metal samples comparable to those of an engineering cutter ring interior. Compared with the engineering cutter ring interior, the base metal samples had better wear resistance and impact toughness but poorer shear strength.

2.Comprehensive performance evaluation

The formula for calculating the comprehensive performance evaluation value *Si* of the quenched samples relative to the engineering cutter ring samples sampled from each region was proposed based on the weighted comparative analysis method:(3)$$S_{i}=\left(-\omega_{i}^{h}S_{i}^{h}+\omega_{i}^{H}S_{i}^{H}+\omega_{i}^{\tau }S_{i}^{\tau }+\omega_{i}^{a}S_{i}^{a}\right)\times 100\%$$
where *i* = 1, 2 and 3, corresponding to the cutter ring edge, the cutter ring interior, and the cutter ring root, respectively; $\omega_{i}^{h}$, $\omega_{i}^{H}$, $\omega_{i}^{\tau }$ and $\omega_{i}^{a}$ represent the weight values (%) of indent depth *h* (μm), hardness *H* (HRC), shear strength *τ* (MPa), and impact toughness *a* (J/cm^2^) when the performance indicators of the quenched samples are compared with those of the engineering cutter ring samples sampled from a given region corresponding to *i*; as shown in Table 3, the weight values are obtained by the expert scoring method; $S_{i}^{h},\;\;S_{i}^{H},\;\;S_{i}^{\tau }$ and $S_{i}^{a}$ represent the scored values of the corresponding performance indicators of the base metal relative to the engineering cutter ring samples sampled from the given region. The scored values can be calculated using the following formula:(4)$$S_{i}^{x}=\left(x-x_{i}\right)/x_{i}$$
where *x* can refer to one of the characters *h*, *H*, *τ*, and *a* which denotes the test value of the corresponding performance index of the base metal samples; *x_i_* represents the test value of the corresponding performance index of the engineering cutter ring samples sampled from the given region corresponding to *i*.

It is worth noting that the negative sign is taken before the $\omega_{i}^{h}S_{i}^{h}\;$ term in formula (3) since the indent depth *d* is inversely proportional to the wear resistance of the cutter. The higher the comprehensive performance evaluation value *S_i_*, the better the comprehensive performance. Based on formula (3), the comprehensive performance evaluation value *S_i_* of the base metal samples obtained in Task 2 was calculated, and then a bar graph was plotted in Figure 18. As can be seen from Figure 18, the comprehensive performance evaluation value *S*_2_ of the base metal samples was equal to 7.6%, which means that the comprehensive performance of the base metal samples was close to that of the engineering cutter ring interior. Therefore, the base metal samples prepared in Task 2 can be used in Task 3 as an economical alternative to the failed engineering cutter rings.

### 4.2. Remanufacturing Performance of the Base Metal

The performance of the surfacing repair layer prepared in Task 3 should be as close as possible to that of the engineering cutter ring edge. To thoroughly investigate the remanufacturing performance of the base metal samples, a single performance evaluation and a comprehensive performance evaluation were carried out.

1.Single performance evaluation

To save space, the performance test results of the remanufactured samples obtained in Task 3 are also summarized in Figure 17. 

As can be seen from Figure 17, the indent depth of the surfacing repair layer was significantly lower than that of the engineering cutter ring samples. However, the hardness, shear strength, and impact toughness of the surfacing repair layer were lower than those of the various parts of the engineering cutter ring. Compared with the physical and mechanical properties of the engineering cutter ring samples sampled from each region, the surfacing repair layer has both advantageous and disadvantageous performance indicators.

2.Microstructure evolution

As shown in Figure 19a, the microstructure of the surfacing repair layer was observed at 100 magnification. As can be seen from the figure, the surfacing repair layer consisted of plate martensite, ferrite, and a small amount of white residual austenite. Due to the high hardness of the plate martensite [23], the surfacing repair layer had a high wear resistance. 

As shown in Figure 19b, the metallurgical bonding layer, which was located between the surfacing repair layer and the base layer (base metal), was observed at 5 magnification. In the figure, the four regions, from left to right, were the base layer, heat-affected zone, and surfacing repair layer. Due to the high heat input into the edge area of the heat-affected zone (near the surfacing repair layer), the temperature within this area can stay above Ac3 (the complete austenitizing temperature) for a long time. During this period, carbide precipitated from the austenite crystal at high temperatures and grew in a needle-like pattern, eventually forming plate martensite after cooling. The temperature of the edge area of the heat-affected zone (near the base layer) did not reach the temperature of phase transformation, so most of the organization in this area was still tempered martensite. 

As shown in Figure 19c, the microstructure of the base layer was observed at 100 magnification. As can be seen from the figure, this region mainly consisted of acicular martensite and residual austenite. The acicular martensite was produced by heat treatment in Task 2. Since the base layer had been quenched but not the surfacing repair layer, the martensite in the base layer was finer than that in the surfacing repair layer. It can explain why the hardness of the base layer was higher than that of the surfacing repair layer in Figure 17. Normally, high hardness leads to excellent wear resistance [20]. Still, the excellent compatibility of the steel ball with the wire material in the indent depth test resulted in a much shallower indent depth in the surfacing repair layer than the base layer. 

3.Comprehensive performance evaluation

Based on formula (3), the comprehensive performance evaluation value *S_i_* of the surfacing repair layer relative to the engineering cutter ring edge was calculated. As shown in Figure 18, *S_i_* was equal to −4.1%, which indicated that the comprehensive performance of the surfacing repair layer was slightly lower than that of the engineering cutter ring edge. Therefore, it would be feasible to employ GMAW to remanufacture the normally-worn ring, but the comprehensive performance of the surfacing repair layer still had room for improvement.

## 5. Conclusions

The simulation and experimental study of the heat treatment process of the H13 steel cutter ring were carried out in this paper. On this basis, considering the actual heat treatment state of the normally-worn ring, the feasibility of surfacing remanufacturing the cutter ring was further evaluated.

The simulation analyses of the heat treatment process route show that the temperature distribution inside and outside was uniform; the microstructure was dominated by martensite, which ensured the high hardness of the base metal samples [24]; the tempering after quenching reduced residual stresses in the remanufactured samples. The simulation results verified the reasonableness of the heat treatment process route set out in this paper, which can be used for subsequent experiments to obtain the base metal samples with excellent comprehensive performance.The analyses of the physical and mechanical properties of the base metal samples show that, compared with the engineering cutter ring interior, the base metal samples had better indent depth, hardness, and impact toughness but poorer shear strength. As the comprehensive performance was similar to the engineering cutter ring interior, the base metal samples can be used to simulate the normally-worn ring.The analyses of the remanufactured properties of the base metal samples show that, compared with the engineering cutter ring edge, the surfacing repair layer had better indent depth but poorer hardness, impact toughness, and shear strength. The comprehensive performance of the surfacing repair layer was slightly lower than that of the cutter ring edge (4.1%), which preliminary verified the possibility of surfacing remanufacturing the normally-worn ring.In the future, the comprehensive performance of the surfacing repair layer would be further improved by optimizing the wire material composition and the surfacing remanufacturing process.

## Figures and Tables

**Figure 1 materials-16-01093-f001:**
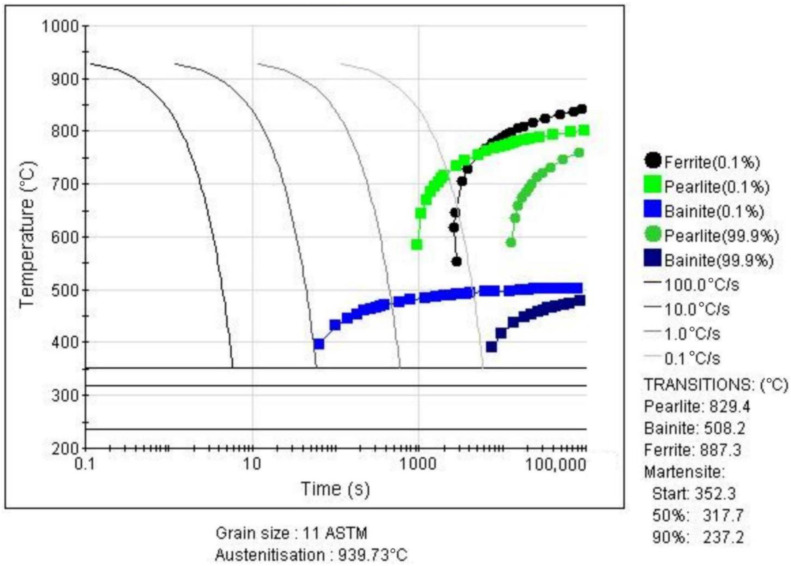
Thermal equilibrium phase diagram of H13 steel.

**Figure 2 materials-16-01093-f002:**
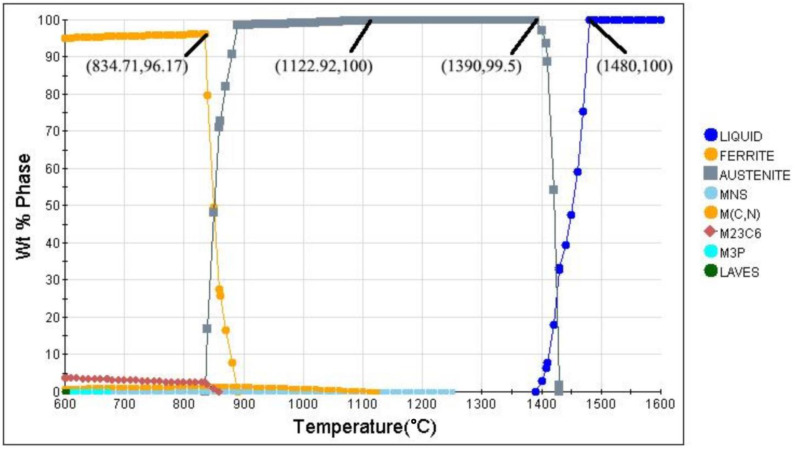
Continuous cooling transition curve of H13 steel.

**Figure 3 materials-16-01093-f003:**
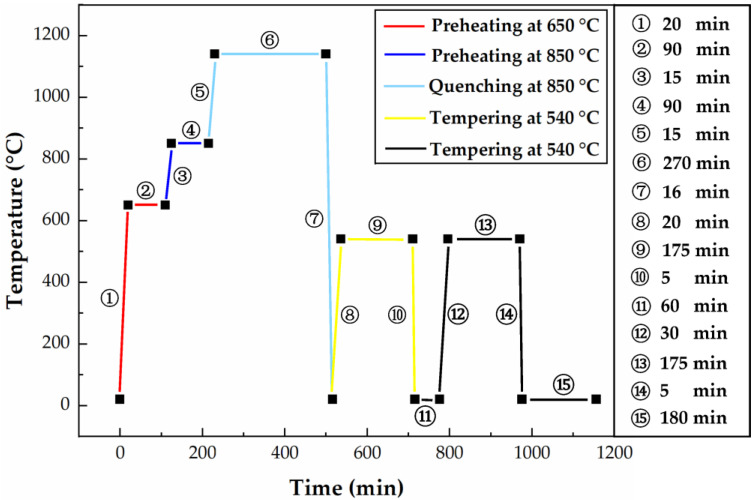
Heat treatment process route of H13 steel cutter ring.

**Figure 4 materials-16-01093-f004:**
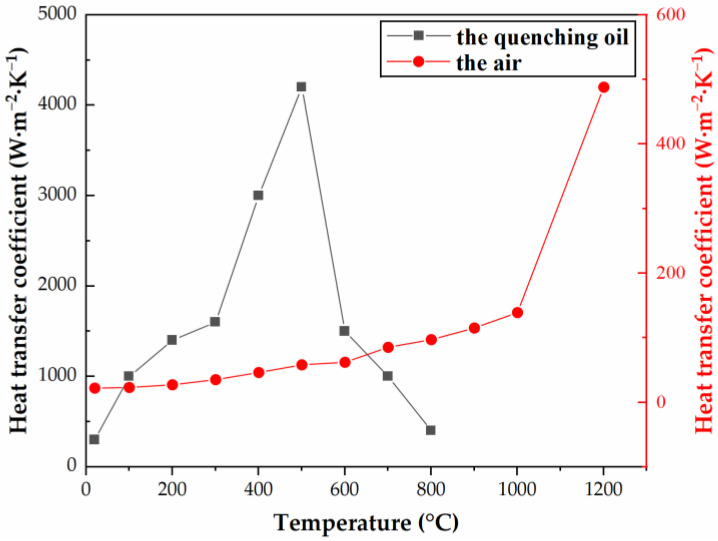
Heat transfer coefficients of the quenching oil and air.

**Figure 5 materials-16-01093-f005:**
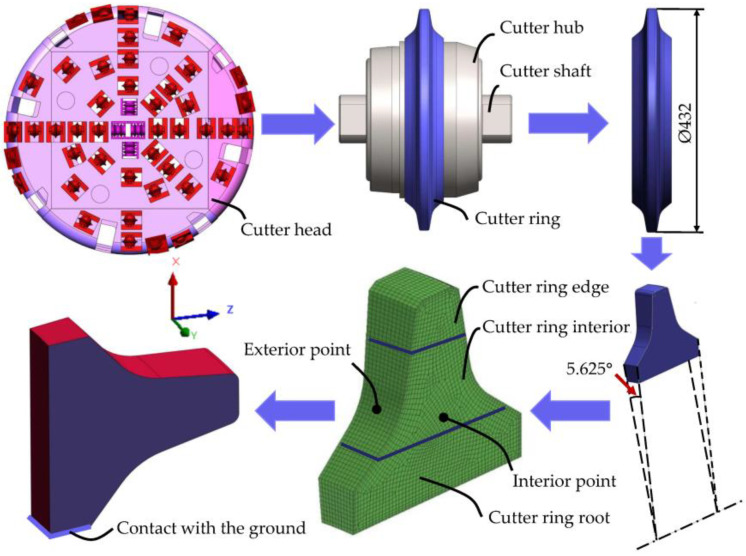
Simplified schematic of FE model of the engineering cutter ring used for simulating the heat treatment process.

**Figure 6 materials-16-01093-f006:**
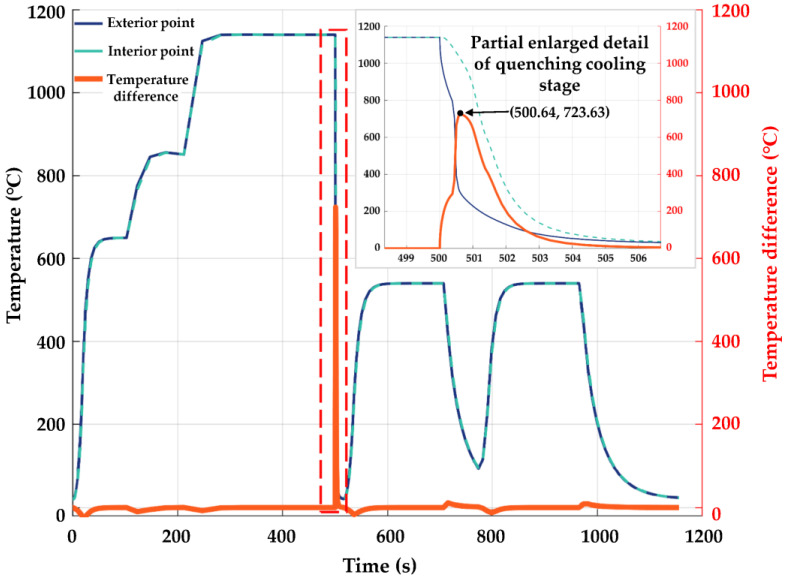
The temperatures at the interior and exterior points and the temperature difference curves between the interior and exterior points.

**Figure 7 materials-16-01093-f007:**
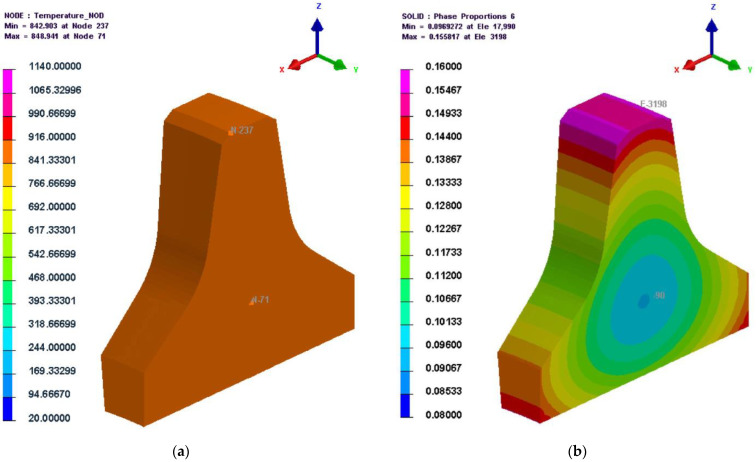
Temperature and microstructure cloud diagrams at 8823 s. (**a**) Temperature; (**b**) Austenite.

**Figure 8 materials-16-01093-f008:**
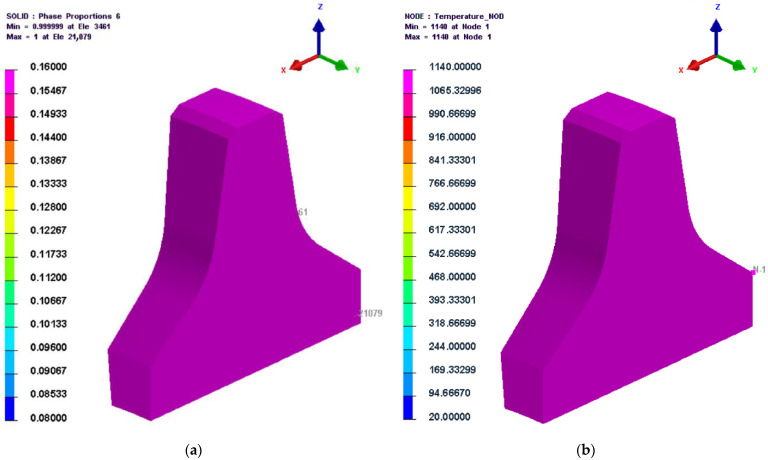
Temperature and microstructure cloud diagrams at 30,000 s. (**a**) Microstructure; (**b**) temperature.

**Figure 9 materials-16-01093-f009:**
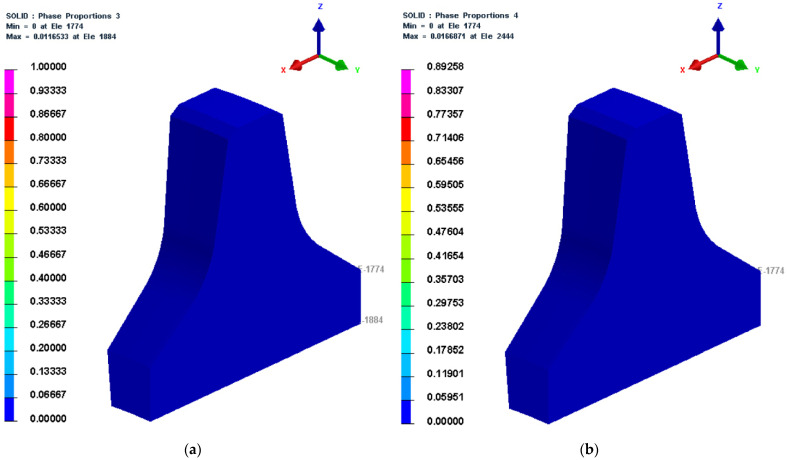
Microstructure cloud diagrams at 30,030 s and 30,033 s. (**a**) Bainite cloud diagram at 30,030 s; (**b**) Martensite cloud diagram at 30,033 s.

**Figure 10 materials-16-01093-f010:**
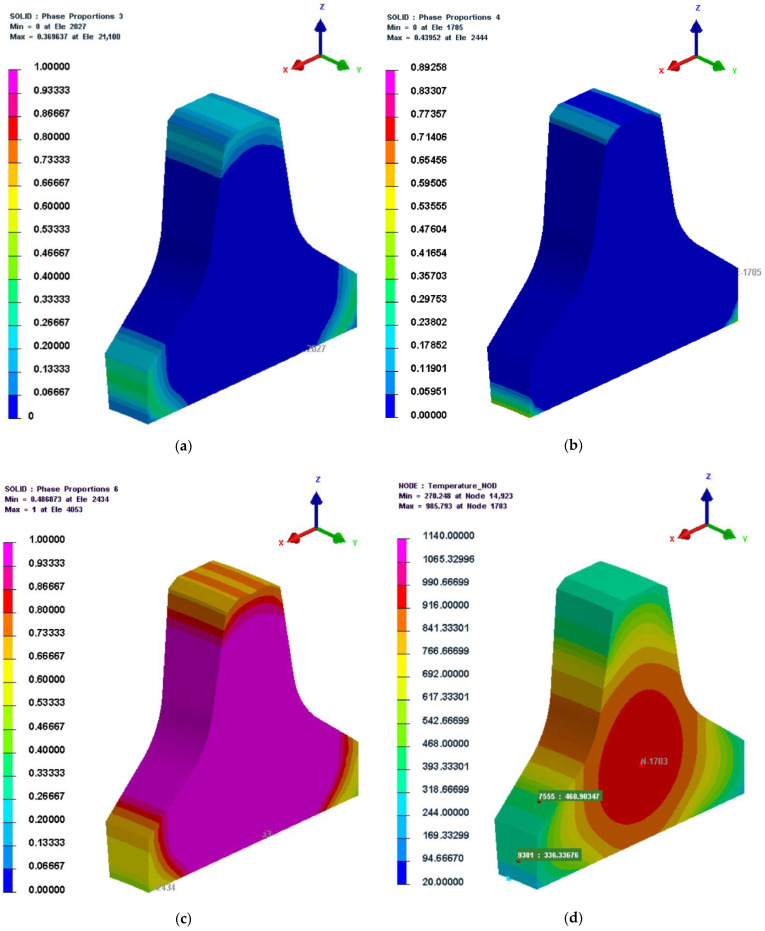
Temperature and microstructure cloud diagrams at 30,045 s. (**a**) Bainite; (**b**) martensite; (**c**) austenite; (**d**) temperature.

**Figure 11 materials-16-01093-f011:**
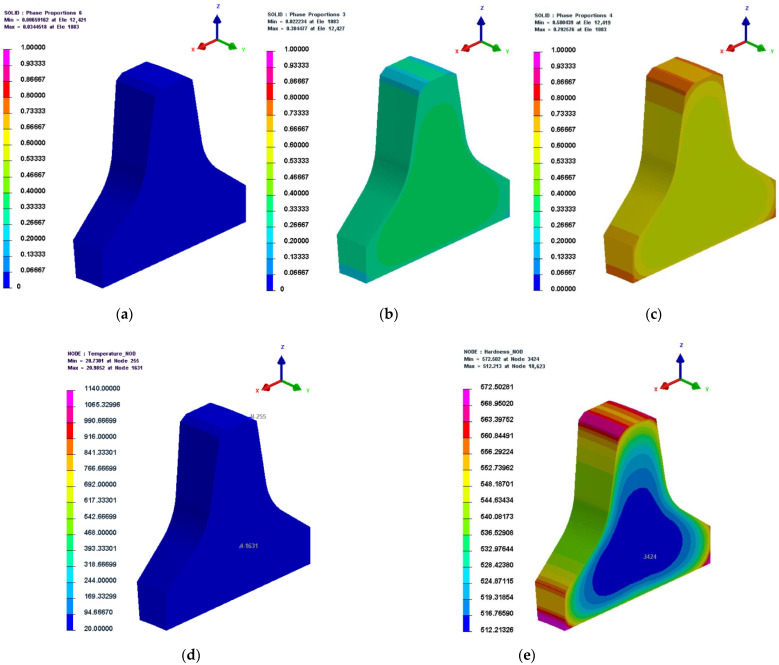
Microstructure, temperature, and hardness cloud diagrams at 30,960 s. (**a**) Austenite; (**b**) bainite; (**c**) martensite; (**d**) temperature; (**e**) hardness.

**Figure 12 materials-16-01093-f012:**
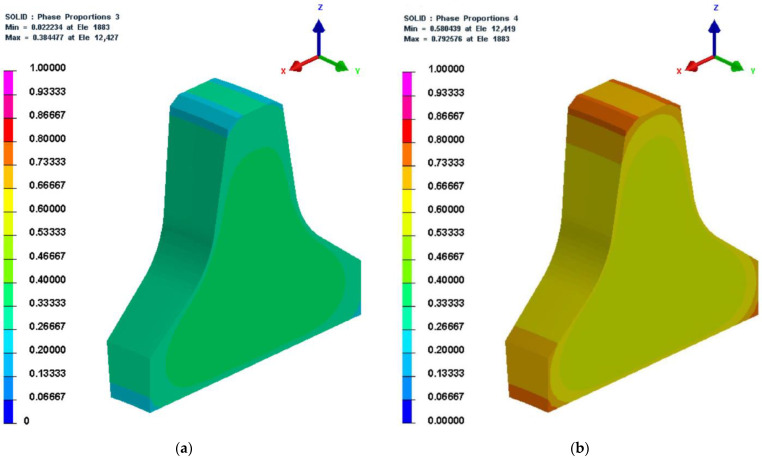
Microstructure, temperature, and hardness cloud diagrams at 69,360 s. (**a**) Bainite; (**b**) martensite; (**c**) temperature; (**d**) hardness.

**Figure 13 materials-16-01093-f013:**
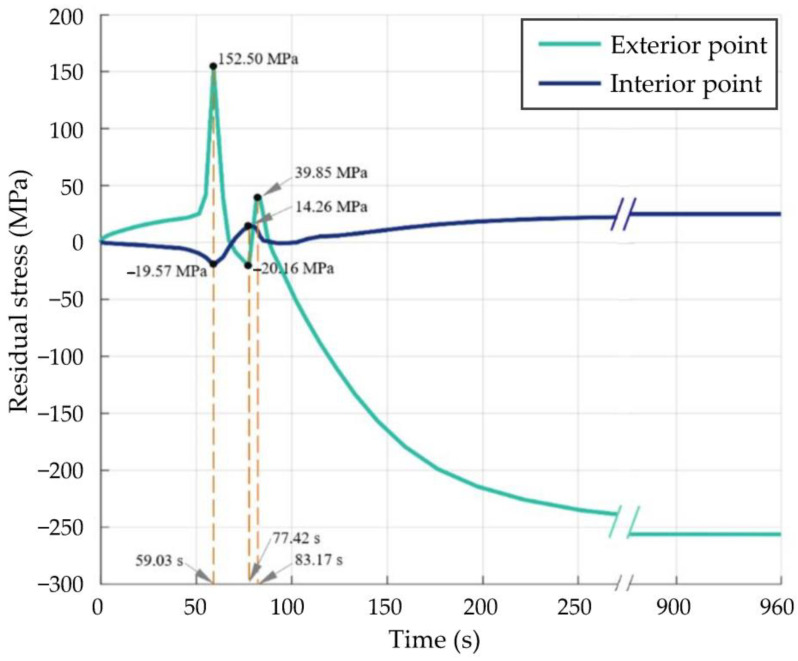
Change curves of residual stress at the interior and exterior points versus during the cooling stages of the quenching process.

**Figure 14 materials-16-01093-f014:**
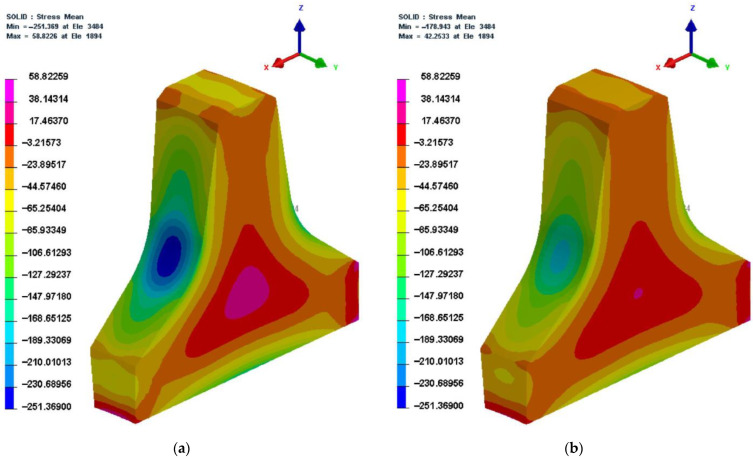
Residual stress cloud diagrams at 30,960 s and 69,360 s. (**a**) at 30,960 s; (**b**) at 69,360 s.

**Figure 15 materials-16-01093-f015:**
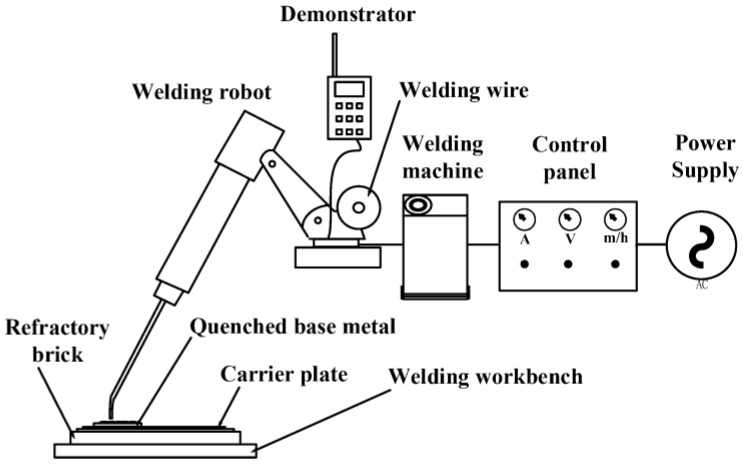
Schematic diagram of the test platform.

**Figure 16 materials-16-01093-f016:**
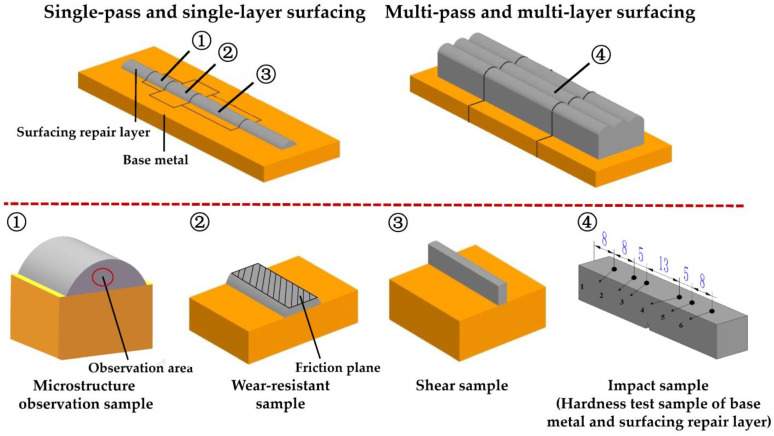
Surfacing methods and sampling for performance tests.

**Figure 17 materials-16-01093-f017:**
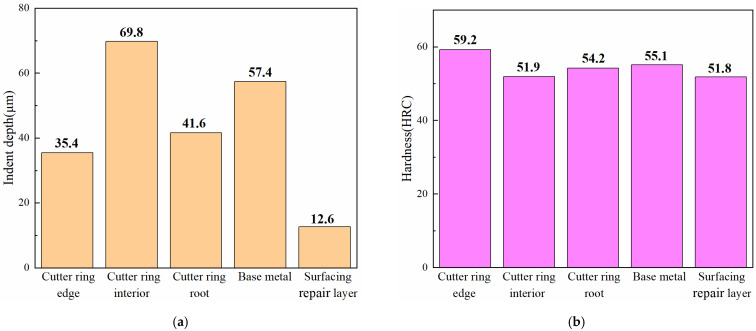
Physical and mechanical properties of the engineering cutter ring samples, the base metal samples, and the surfacing repair layer samples. (**a**) Indent depth; (**b**) hardness; (**c**) shear strength; (**d**) impact toughness.

**Figure 18 materials-16-01093-f018:**
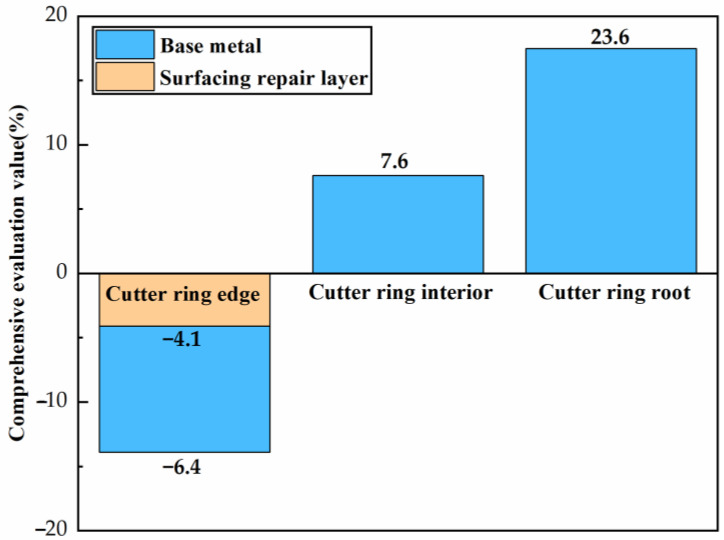
Bar chart of the comprehensive performance evaluation value of the base metal samples and the surfacing repair layer samples.

**Figure 19 materials-16-01093-f019:**
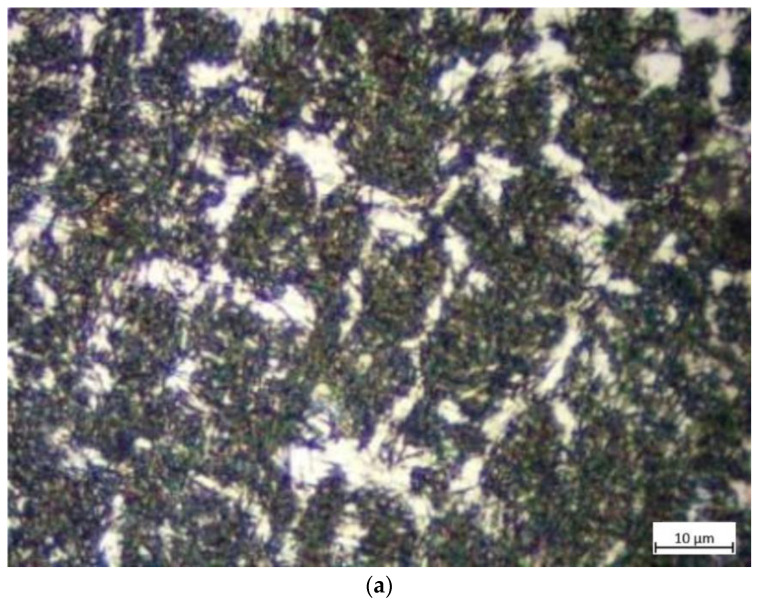
Microstructure of the remanufactured samples. (**a**) Surfacing repair layer; (**b**) bonding layer; (**c**) base layer.

**Table 1 materials-16-01093-t001:** Chemical composition of the base metal, the engineering cutter ring, and flux-cored wire (Wt. %).

	C	Mn	Cr	Mo	Si	V	P	S
base metal	0.342	0.31	4.82	1.24	0.83	1.11	≤0.03	≤0.03
engineering cutter ring	0.33	0.32	5.3	1.68	0.95	0.93	≤0.03	≤0.03
flux-cored wire	0.388	0.387	5.01	1.25	0.93	0.95	≤0.03	≤0.03

**Table 2 materials-16-01093-t002:** Performance test projects and the test instruments for each task.

Test Project	Task 1	Task 2	Task 3	Testing Instrument
Microstructure	×	×	√	MR2100 metallographic microscope
Indent depth	√	√	√	CFT-I friction tester, VHX-2000C super depth-of-field microscope
Shear strength	√	√	√	WAW-300 universal testing machine
Impact toughness	√	√	√	JB-500B Charpy pendulum impact machine
Hardness	√	√	√	HR150-A rockwell apparatus

**Table 3 materials-16-01093-t003:** The weight values of physical and mechanical properties of the quenched samples relative to the engineering cutter ring samples sampled from each region (%).

	Indent Depth $\omega_{i}^{h}$	Hardness $\omega_{i}^{H}$	Shear Strength $\omega_{i}^{\tau }$	Impact Toughness $\omega_{i}^{a}$
Edge (*i* = 1)	30	30	20	20
Interior (*i* = 2)	20	20	30	30
Root (*i* = 3)	10	20	40	30

## Data Availability

Data sharing is not applicable to this article.

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
