# Peer review of "Feasibility Study for the Remanufacturing of H13 Steel Heat-Treated TBM Disc Cutter Rings with Uniform Wear Failure Using GMAW"

_materials, 2023, doi:10.3390/ma16031093_

Round 1

Reviewer 1 Report

The article is a extensive deep research, but there are several comments:

1. The Introduction should describe the recovery technology used in terms of frequency of use and applicability in different industries, complexity, labor intensity, economic efficiency, etc.

2. Figure 3 is overloaded with inscriptions. It is intuitively clear which inscription belongs to which line, but it is perceived a little complicated. Perhaps it is better to designate the segments with letters or numbers and indicate in the legend or caption the time of the exposure process at the specified temperatures.

3. The results are discussed in section 4, not in section 5. Therefore, the title of section 4 should be changed to "Experimental Results and Discussion", and section 5 should be renamed to "Conclusions".

Reviewer 2 Report

Journal: Materials (ISSN 1996-1944)

Manuscript ID: Materials- 2167609

The authors presented an article on “Feasibility Study for the Remanufacturing of H13 Steel Heat-Treated TBM Disc Cutter Rings with Uniform Wear Fail-ure Using GMAW”. I think the article is well organized and suitable for the "Materials" journal. But the article will be ready for publication after a major revision. Comments are listed below.

1.      More cites are needed in the introduction. The usage areas and superior aspects of TBM disc cutters should be supported with examples from the literature.

2.      In the introduction, the heat treatment that causes the improvement of the mechanical properties should be further detailed.

3.      The authors mention many performance evaluation indices for TBM breakers. What are these indices? (Page 2, line 92).

4.      In the last paragraph of the introduction, the prominent aspects of the study and its differences from other studies should be clearly stated.

5.      The authors stated that the workpieces were cooled with oil during the quenching process and air cooled during the tempering process. Why did they use two different cooling? It should be explained (Page 24, line 153).

6.      Is the unit "mm" correct in the sentence "If the radial wear of the cutter ring reaches 35 mm during engineering maintenance" on page 18, line 408?

7.      The results section is devoid of discussion. It should be supported by the literature.

8.      The relationships between different test results (Indent depth, Hardness, Shear strength, Impact toughness) should be discussed.

9.      Scala is not read in Figure 19. It should be enlarged.

10.  The article contains numerous typographic and language errors. It should be corrected.

11.  The article should be rearranged by taking into account the journal writing rules and citation rules.

12.  The paper is well-organized, yet there is a reference problem. First, your reference list contains no article from the “Materials” journal. If your work is convenient for this journal's context, then there are many references from this journal. Secondly, cited sources should be primary ones. Namely, the indexed area shows the power of a paper and directly your paper's reliability. Please make regulations in this direction.

*** Authors must consider them properly before submitting the revised manuscript. A point-by-point reply is required when the revised files are submitted.

Reviewer 3 Report

- What was your most important innovation in this article? Please state it clearly in the text of the article.

- In the title of the article, it should be specified whether the research is based on numerical simulation or experimental work?

- The considered conditions for the numerical simulations are incomplete, and full explanations regarding the type of problem solving, boundary conditions, and the Yendi network must be provided.

- Authors should explain how and by what method they confirmed their numerical simulations?

- The relationship between the numerical and experimental results should be presented more accurately and practically.

- The explanations given about the residual stress changes in Figure 13 are not complete and convincing.

- The authors should explain how they obtained the microstructural changes from Figure 12. In my opinion, the given figures do not represent the microstructure.

Round 2

Reviewer 2 Report

Journal: Materials (ISSN 1996-1944)

Manuscript ID: Materials- 2167609

 Round 2#

The authors made the desired corrections. In my opinion, this article can be accepted for publication in the "Materials" journal in its final form.

Reviewer 3 Report

The manuscript can be accepted in the present form.